# Insights into the Function of Regulatory RNAs in Bacteria and Archaea

Elahe Soltani-Fard [1,†], Sina Taghvimi [2,†], Zahra Abedi Kichi [3,4,†], Christian Weber [3], Zahra Shabaninejad [5], Mortaza Taheri-Anganeh [6], Seyyed Hossein Khatami [7], Pegah Mousavi [8], Ahmad Movahedpour [6,*] and Lucia Natarelli [3,*]

1   Department of Molecular Medicine, School of Advanced Technologies, Shahrekord University of Medical Sciences, Shahrekord 8815713471, Iran; elahesoltanifard69@gmail.com
2   Department of Biology, Faculty of Science, Shahid Chamran University of Ahvaz, Ahvaz 6135743135, Iran; SinaTaghvimi@yahoo.com
3   Institute for Cardiovascular Prevention (IPEK), Ludwig-Maximillians University, 80336 Munich, Germany; Zahra.Abedi@med.uni-muenchen.de (Z.A.K.); christian.weber@med.uni-muenchen.de (C.W.)
4   Department of Genetics, Faculty of Biological Sciences, Tarbiat Modares University, Tehran 14115-154, Iran
5   Department of Nanobiotechnology, Faculty of Biological Sciences, Tarbiat Modares University, Tehran 14115-154, Iran; shabanizahra1369@gmail.com
6   Department of Medical Biotechnology, School of Advanced Medical Sciences and Technologies, Shiraz University of Medical Sciences, Shiraz 71345-1583, Iran; mortazataheri@yahoo.com
7   Department of Clinical Biochemistry, School of Medicine, Shahid Beheshti University of Medical Sciences, Tehran 1985717443, Iran; shossein.khatami@gmail.com
8   Department of Medical Genetics, Faculty of Medicine, Hormozgan University of Medical Sciences, Bandar Abbas 7919693116, Iran; pegahmousavi2017@gmail.com
*   Correspondence: ahmad.movahed14@gmail.com (A.M.); lnatarel@med.lmu.de (L.N.)
†   Authors equally contribute at the manuscript.

**Abstract:** Non-coding RNAs (ncRNAs) are functional RNA molecules that comprise about 80% of both mammals and prokaryotes genomes. Recent studies have identified a large number of small regulatory RNAs in *Escherichia coli* and other bacteria. In prokaryotes, RNA regulators are a diverse group of molecules that modulate a wide range of physiological responses through a variety of mechanisms. Similar to eukaryotes, bacterial microRNAs are an important class of ncRNAs that play an important role in the development and secretion of proteins and in the regulation of gene expression. Similarly, riboswitches are *cis*-regulatory structured RNA elements capable of directly controlling the expression of downstream genes in response to small molecule ligands. As a result, riboswitches detect and respond to the availability of various metabolic changes within cells. The most extensive and most widely studied set of small RNA regulators act through base pairing with RNAs. These types of RNAs are vital for prokaryotic life, activating or suppressing important physiological processes by modifying transcription or translation. The majority of these small RNAs control responses to changes in environmental conditions. Finally, clustered regularly interspaced short palindromic repeat (CRISPR) RNAs, a newly discovered RNA regulator group, contains short regions of homology to bacteriophage and plasmid sequences that bacteria use to splice phage DNA as a defense mechanism. The detailed mechanism is still unknown but devoted to target homologous foreign DNAs. Here, we review the known mechanisms and roles of non-coding regulatory RNAs, with particular attention to riboswitches and their functions, briefly introducing translational applications of CRISPR RNAs in mammals.

**Keywords:** non-coding RNA (ncRNA); microRNA; riboswitches; CRISPR; regulatory RNA

## 1. Introduction

The cell regulatory network is a complex set of molecular factors that interact with each other and genes, thus controlling gene expression. In the simplest case, transcription

factors bind to regulatory sequences, and the result of this binding can be to induce or inhibit the expression of genes [1]. On the other hand, with the discovery of RNA in 1967, it was found that this molecule plays a key role in all types of life, and, on average, 20% of the dry weight of prokaryotic cells is characterized by a genome codifying for RNAs [2,3]. The investigation of RNAs is dependent on their recognition as both an inherited genetic information interface and a chemical catalyst. In addition, changes in the environment are inevitable, and RNAs play an essential role in enduring this critical event [4]. Regulatory RNAs in prokaryotes are a heterogeneous group of molecules that, by multiple mechanisms, induce different physiological responses [5]. Recent molecular biology studies show that non-coding RNAs (ncRNAs) finely regulate most of the cellular regulatory functions. Researches in the last decade have confirmed the role of these ncRNAs in various processes [6]. Based on their constitutive pair base, ncRNAs are divided into two main groups: small and large non-coding RNAs (sncRNAs and lncRNAs, respectively). The sncRNAs subclass includes two well-known functional RNAs, known as transfer and ribosomal RNAs, and two regulatory RNAs, known as microRNAs (miRNAs) and small interfering RNAs (siRNAs), which are involved in the regulation of gene expression. Overall, so far, only about 100 small RNAs have been experimentally validated [7]. Unlike sncRNAs, lncRNAs are a heterogeneous group of RNAs longer than 200 nucleotides (nt) with an intricate network of loops and bulges, of which the function in prokaryotes is largely unknown [8].

In recent years, extensive research identified a variety of ncRNAs. Some of these ncRNAs, such as miRNAs, which act as ribonucleoprotein complexes that suppress transcription, and clustered regularly interspaced short palindromic repeat (CRISPR) RNAs, which use splicing phage DNA as a defense mechanism for the bacterium [9]. The most comprehensive studies have been conducted on small RNAs, and it has been shown that these regulators act by linking nucleotide bases to RNA and alter RNA stability and translation processes [2]. Other types of ncRNAs have a protein-independent function [9]. Riboswitches are a specific example of this category that can control gene expression. Riboswitches sense and respond to the availability of different nutrients in cells, thus modulating metabolic pathways [2]. These RNAs were first described 10 years ago and recognized as essential factors in controlling gene expression in a wide range of bacteria. Riboswitches are usually classified as metabolite-sensing RNAs, which are embedded within the 5′-untranslated region (UTR) of their mRNAs, and they respond to different environmental changes by altering gene expression and by binding to small molecules [4]. These molecules are metabolite-sensing RNAs embedded with their mRNA at their 5′-UTR and respond to different environmental changes by altering gene expression [9]. This review will summarize the genetic structure of prokaryotes and bacterial regulatory RNAs, focusing on small RNAs and riboswitches.

## 2. Genes and mRNAs Structures in Prokaryotes

Prokaryotes, which include bacteria and archaea, have smaller genomes than eukaryotes [10,11]. The prokaryotic chromosome is generally assumed to consist of a circular, double-stranded DNA chromosome with a less extensive coiling structure than eukaryotic chromosomes [12]. Genome sequencing data recently revealed that 10% of prokaryotes contain several essential and large replicons of either circular or linear DNA, named "chromid" [13]. Hence, the prokaryotic genome architecture is currently referred to as a multipartite genome, which comprises a chromosome and more additional large replicons. The prokaryotic genome size ranges from around 50 Kb to more than 13 Mb [14,15]. The multipartite genome comprises differences in codon usage, and it is usually very compact, with a gene density typically approaching 85% [16]. The chromosomes of bacteria are organized into domains, characterized by supercoiled loops independent from one another [12]. Bacterial chromosomes (referring to as "primary replicon" according to the current nomenclature) are generally 1000 times longer than the cells from which they belong to [17].

Prokaryotic chromatin is dense and forms a pseudo-compartment that frequently occupies a distinct region within the cell, and it is distinguished by the absence of ribosomes [18]. Nucleoid refers to the dense chromatin area that is functionally equivalent to the eukaryotic nucleus [19]. Unlike eukaryotes, prokaryotic cells typically lack a nucleus-limiting membrane and a clear physical separation between DNA and the cytoplasm [18,20]. The chromosome is always the largest replicon in the genome, comprising the bulk of the core/essential genes, and does not have the intron-exon arrangement that eukaryotic genes have (with occasional exceptions). Altogether, these factors facilitate prokaryotic genome studies compared to the more complex eukaryotic genome [21]. The organization of chromosomal genes partly reflects functional and regulatory purposes. Notably, the chromosomal locations of genes involved in catabolism and biosynthesis are usually encoded by a single operon and colocalize together with their regulators [22]. Secondary replicons usually lack core genes and are, therefore, dispensable for cell viability. A class of secondary replicons comprises small plasmids and chromids, which show differences in their sizes and regulatory mechanisms. Unlikely plasmids, chromids carry at least one core-essential gene, leading to a classification of chromides as an intermediate between plasmids and chromosomes, showing a comparable or slightly lower number of core genes [23].

The prokaryotic chromosome is replicated by a single replication bubble ("unibubble" replication), reflected in the familiar Cairns structure of theta-replication. The genome contains a variety of genes, including those encoding for ribosomal RNAs (rRNAs), transfer RNAs (tRNA), and other noncoding RNA genes, in addition to protein-coding genes. All these sub-classes of RNAs show a variably complex secondary structure, reflected in their broad range of functions. Indeed, all small RNAs regulate several post-transcriptional events, such as splicing [24], subcellular localization [25], translation [26], and decay [27] of several RNA transcripts. A 3′-UTR is found downstream of the translational stop codon in most prokaryotic mRNAs and contains intrinsic, rho-independent, transcription termination stem-loops [28]. Compared to eukaryotes mRNAs that exhibit a half-life within the range of an hour, bacterial mRNAs exhibit an half-life of only few minutes, a considerably shorter time due to their high instability in vivo [29].

The fundamental genetic processes of DNA replication and transcription are required for prokaryotic growth and division. They are carried out by large protein complexes that move quickly and over long distances along the chromosomes [30,31]. Bacterial DNA replication and transcription processes rely on the same template and occur concurrently. Prokaryotic open reading frames (ORFs) are often organized into a polycistronic operon under the control of a shared set of regulatory sequences [32]. Polycistronic mRNAs carry the information of several protein-coding genes, subsequently translated into several proteins. Multiple ribosomes immediately translate nascent transcripts. Notably, membrane proteins-related transcripts are simultaneously translated as multiple protein complexes from a unique polycistronic mRNA transcript with the same operon and the same related function; this is quite a prerogative of prokaryotes. Hence, transcription, translation, and protein localization are tightly linked. Gene expression is high in exponentially growing cells. Therefore, collisions between the fast replication fork and the slower RNA polymerases are common [20]. In prokaryotes, the protein synthesis machinery has a direct access to mRNAs and occurs immediately after it is transcribed from the genome.

Since several genes can be transcribed in a single polycistronic mRNA, and because of transcripts short half-life, bacterial mRNAs are used very rapidly and effectively [33]. Therefore, it requires fine-tune regulatory mechanisms that rely on transcription-to-translation closed linkage, such as small RNAs [34].

## 3. miRNA-Size Molecules and Small RNAs in Prokaryotes

As mentioned before, small prokaryotic RNAs can be classified according to three features: (i) their size, ranging between 50 and 500 nt in length; (ii) their subsequent—high complex—structural diversity; and (iii) their mechanism of action. Hence, we will discuss the regulatory small RNA sub-classes by comparing prokaryotic small RNAs

with their respective eukaryotic-like homologs, such as prokaryote miRNA-size molecules (eukaryotic miRNA-like), and properly classified small RNAs, such as *trans*-encoded small RNAs (eukaryotic siRNA-like), and *cis*-encoded small RNAs (eukaryotic antisense small RNA-like and miRNA-like) [35].

### 3.1. miRNA-Size Molecules

Despite the thousands of genes identified in the human genome encode for proteins, as messenger RNAs, approximately 95% of the genome encode for non-coding molecules, and the percentage varies between species [36,37]. An important subclass of ncRNA molecules with a known regulatory function includes miRNAs, representing one of the three main types of all prokaryotic ncRNAs. MiRNAs are 21 to 25 nt in length [38–40] and regulates gene expression primarily by acting as post-transcriptional repressors [35,37]. miRNAs were first described in 1993 by Ambros and colleagues in *Caenorhabditis elegans* (*C. elegans*) as RNA molecules of 18 to 23 nt in length that regulate developmental timing [41–46]. Recently, several miRNA-size small RNA fragments with 15–26 nt in length have been identified in prokaryotes, such as *Streptococcus mutans*, *Escherichia coli* (*E. coli*), and *Mycobacterium marinum*. Moreover, they have been described also in periodontal pathogens, but their mechanism of biogenesis has not yet been determined. In *Mycobacterium marinum*, a small RNA of 23 nt in length, named MM-H, has been classified as miRNA-like small RNA and, similar to miRNAs, is composed of a precursor stem-loop structure that requires the eukaryotic host machinery to be cleaved [47]. Hence, bacterial miRNAs may affect host gene expression rather than bacterial genes. A recent work from Nejman-Faleńczyk and colleagues reported the existence of a phage-derived miRNA-size molecule with 20-nt in length isolated from infected *E. coli*, named 24B_1, encoded in the lom-vb_24B_43 region of the phage Φ24B genome, and produced by the cleavage of a longer (80-nt long) transcript. As promoter of phage lysogeny.24B_1 is the first example of a phage miRNA-size molecule having a proved a physiological significance in bacterial cells [48]. Next-generation sequencing (NGS) consents to identify approximately 24 to 30 additional RNAs categorized as miRNA-size. Twenty-four hairpin-structured precursors are validated prokaryotic miRNA-size molecules, but their exact cleavage sites for the final processing into mature miRNA-size have not been confirmed. However, NGS, northern blot, and expression analysis revealed the existence of a fixed-length form for some of them that would suggest a selective cutting system from a longer precursor. Hence, considering that hairpin structures are crucial for the processing of eukaryotic miRNAs, these prokaryotic miRNA-size molecules might be considered as miRNAs-type molecules, and similar to eukaryotic miRNA maturation steps, prokaryotic miRNAs-size molecules might undergo a similar multi-step maturation process. Indeed, in eukaryotes, RNA polymerase II-mediated transcription of miRNA genes generates a hairpin-like RNA named pri-miRNA, with a length of 1 to 3 kb. Pri-miRNAs are cleaved in the nucleus by the RNaseII enzyme Drosha, which generates ~70 nt precursors called pre-miRNAs. Then, pre-miRNAs are released from the nucleus to the cytoplasm by exportin-5 to proceed with the last maturation step. In the cytoplasm, RNase II enzyme Dicer converts pre-miRNAs into mature miRNA duplexes of 20-nt long. Dicer is regulated by various factors, and interestingly, it can be inhibited by miRNAs through a self-regulatory mechanism in humans [39]. Recent findings indicate that similarly to eukaryotic Dicer, a prokaryotic Dicer-like enzyme named MazF may play a similar role in *E. coli* within the maturation of miRNA-size transcripts [49]. Notably, one strand of the eukaryote miRNA duplex is usually degraded, termed "passenger strand", whereas the "guide strand" is retained or loaded in the RNA-induced silencing complex (RISC), following the binding to the RISC-member argonaute protein 2 (AGO2). However, recent findings indicate that both strands of the miRNA duplex can be conserved and regulate distinct transcript targets, at least in mammals [50]. miRNA guide strands usually exert an inhibitory role by targeting the 3′-UTR of their mRNA targets with their seed sequence [50].

Like eukaryotic Argonaute proteins, AGO-encoding genes have been identified in about one-tenth of the known bacteria species. Some of them, including Argonautes from *Marinitoga piezophila* (MpAgo) and *Rhodobacter sphaeroides* (RsAgo), use small guide RNAs, similar to their eukaryotic counterparts. However; the function, the molecular mechanisms, as well as the target pathways of these Argonaut-like enzymes remain largely unknown [51]. While miRNAs have long been thought to play only an inhibitory role in the translational process, recent evidence indicates that under some circumstances, they can actually promote translation. miRNAs may often cause the target mRNA to be lost [52]. Those miRNAs that are fully coupled to the target mRNA induce endonuclease cleavage. During this process, the molecule is completely destroyed due to the conserved parts' loss at the mRNA's start and end [53].

Some prokaryotic miRNA-size molecules are also protected from degradation and might play a role in prokaryote-to-prokaryote or prokaryote-to-eukaryote communication. The ability of miRNAs to degrade mRNAs has two important consequences. First, by decreasing the number of mRNAs, the efficiency of mRNA production is drastically reduced. Second, the inhibitory effect of miRNA through mRNA degradation irreversibly inhibits protein expression [54]. One of the roles of prokaryotic miRNAs is to interfere with cell evolution. For example, two miRNAs, named lin-4 and let-7, identified in *C. elegans* are involved in larval development control and timing [55]. Moreover, these RNAs play a role in stem cell differentiation as well, such as miRNA 296, of which the levels decrease, and miR21/22, of which the levels increase, during stem cell differentiation [56].

Prokaryotic miRNAs can also accumulate in the nucleolus as pre-miRNAs and mature miRNAs. There are many hypotheses regarding the function of these neucleolus-enriched miRNAs. For example, miR-206 localization loci are identical to that of 28S rRNA in the nucleolus and cytoplasm of mammalian cells, suggesting that this miRNA might be involved in the early stages of ribosome biogenesis [57,58].

### 3.2. Small RNAs

The ability of bacteria to sense and adapt to environmental changes relies on their organized and timely control of gene expression, especially in highly variable and sometimes stressful circumstances. As a result, bacteria have developed a variety of pathways for regulating gene expression in response to environmental cues, which necessitates the integration of external signals as well as the coordination of internal responses. Bacteria control gene expression using various small regulatory RNAs [59,60]. Small RNAs were first discovered in bacteria in the early 1970s, but their role was better investigated only in the last decades through genome-wide interaction analyses that revealed how small RNAs are related to mRNA target genes [61–64]. Only 100 of all predicted small RNAs have been widely validated in *E. coli*. These small RNAs are encoded by RNA genes in chromosomal intergenic regions from Gram-positive and Gram-negative bacteria, as well as in obligate intracellular rickettsia and spirochetes [65–67]. The prokaryotic small RNAs vary greatly in size. Typically, they are 50 and 500 nt in length and commonly do not contain ORFs. Small RNAs play key roles in a number of prokaryotic physiological responses, including the control of carbon metabolism, virulence, motility, biofilm production, bacterial adaptation to environmental changes, and stress response [68,69], by modulating prokaryotic transcription and translation. Individual small RNAs show different mechanisms of action, including the sequestration of regulatory proteins [70,71] and the base pairing with target mRNA molecules to increase or decrease their stability and/or translation [72,73]. Base pairing complementarity is the one of main criteria by which prokaryotic small RNAs can be categorized in *cis*-encoded and *trans*-encoded small RNAs (Figure 1).

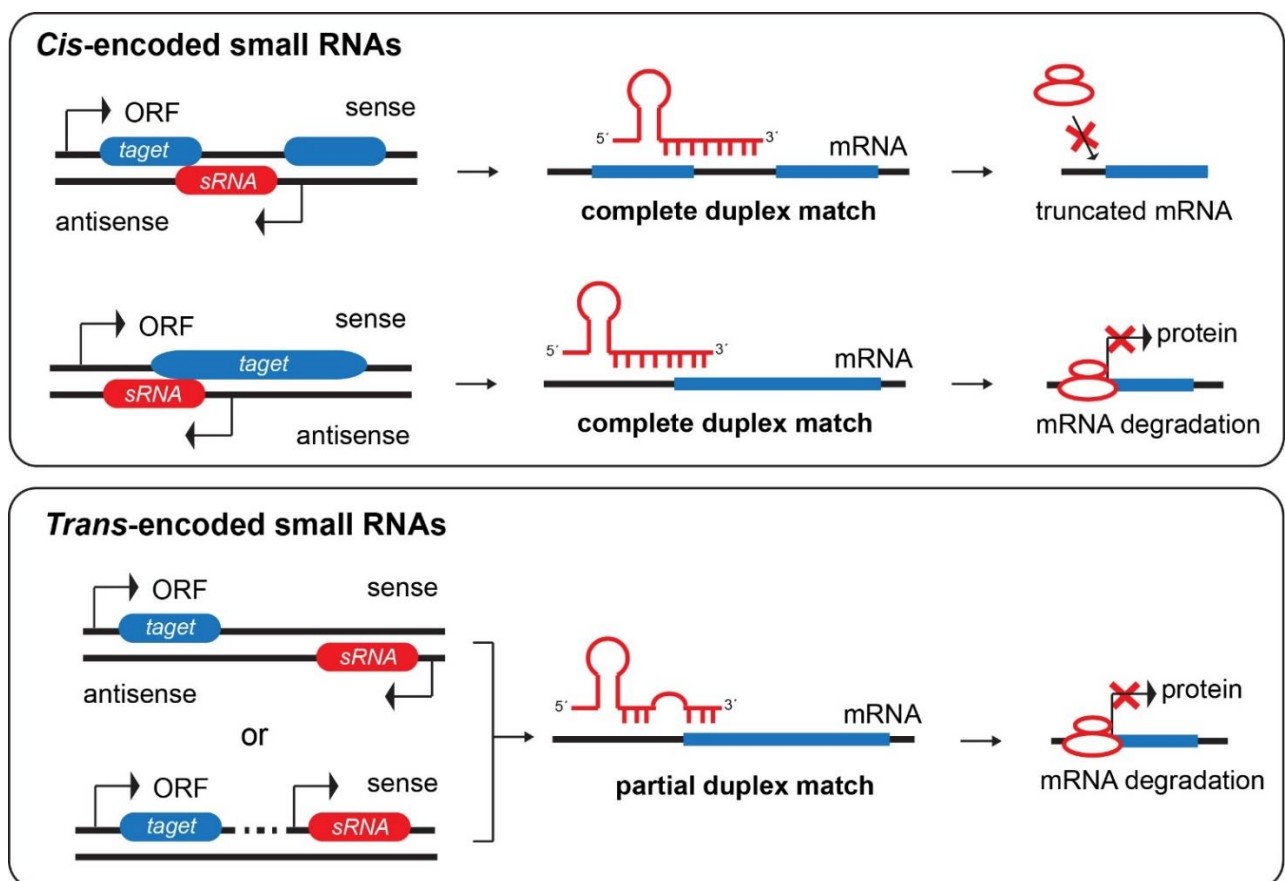

**Figure 1.** *Cis*-encoded small RNAs are transcribed from the complementary strand of the prokaryotic genome and from the same genetic loci of their target genes. They show high complementarity with their mRNA targets and regulate the maintenance and the stability of RNA transcripts, by inhibiting translational repression or by inducing RNA degradation and cleavage. *Trans*-encoded small RNAs are transcribed from DNA sense and antisense genes, from different genomic locations compared to their target genes. They show partial complementarity with their target genes. Their binding to mRNA targets inhibits protein translation or induces mRNA degradation.

### 3.2.1. *Cis*-Encoded Small RNAs

*Cis*-encoded small RNAs are the most common type of antisense RNAs in small prokaryotes. They are transcribed from the complementary (opposite) strand of prokaryotic genome and from the same genetic locus of their target genes. As a result, *cis*-encoded small RNAs often have extensive antisense complementarity (>75 nt) with their target mRNAs (Figure 1). *Cis*-encoded small RNAs comprise a complex secondary structure with a long stem and high complementary loop. Their high complementarity with their mRNA targets minimizes non-selective target recognition and increases their action efficiency. They regulate the maintenance and stability of mobile genetic elements by inhibiting primer maturation, transcriptional attenuation, and translational repression or inducing RNA degradation and cleavage (Figure 1) [74].

One group of *cis*-encoded antisense small RNAs modulates gene expression in a possible operon, which is usually between two genes (Figure 1). Some of these small RNAs are encoded in regions complementary to the intervening sequence between ORFs. For example, the base pairing between the stationary phase-induced GadY *cis*-encoded antisense small RNA and the gadXW mRNA causes the duplex between the gadX and gadW genes, which in turn is cleaved and results in an increase in the gadX transcript level in *E. coli* [75,76]. An additional example of *cis*-encoded antisense small RNAs comes from iron stress-repressed RNAs (IsrRs), which are antisense RNA transcribed from the *isiA* non-coding strand. isiA is a known protein that accumulates during iron starvation and

oxidative stress. IsrR is present in iron-rich environments but shows an inverse correlation with the levels of iron in bacteria. Therefore, IsrR filters out environmental iron stress signals through the suppression of isiA expression in *Synechocystis* [77].

The so-called type I toxin-antitoxin (TA) systems contain many *cis*-encoded small RNAs [78]. There are two-gene components that contain a stable protein toxin and an unstable *cis*-encoded small RNA antitoxin that base-pairs with toxin mRNA, avoiding translation and inducing degradation [79,80]. While high levels of toxins destroy cells, lower levels produced by single-copy loci under inducing conditions can only delay cell growth. Despite the fact that the mechanism is unknown, one hypothesis is that chromosomal toxin-antitoxin modules induce slow growth or stasis in stressed cells to allow them to repair damage or to adapt to environmental changes [81,82]. Another proposed theory is that such modules are retained in bacterial chromosomes as a defense against plasmids containing homologous modules, suggesting that the chromosomal antisense small RNA can suppress the expression of plasmid-encoded toxin.

### 3.2.2. *Trans*-Encoded Small RNAs

*Trans*-encoded small RNAs are a big group of prokaryotic small RNAs that share similarities with both eukaryotic miRNAs and small interfering RNAs (siRNAs), with some crucial differences. *Trans*-encoded small RNAs are about 20 to 30 nt in length, such as eukaryotic miRNAs, produced from DNA sense and antisense genes, pseudogenes, and inverted repeats (Figure 1) [7,36]. Unlike eukaryotic miRNAs, *trans*-encoded small RNAs are not processed from an precursor with approximately 100 nt in length in a shorter fragment of 20 nt long, but they are rather transcribed as a single mature transcript. Prokaryotic small RNAs derived from a maturation step of the longer transcript have been anyway described, but they are still longer than their respective eukaryotic siRNAs and miRNAs. Their stability is guaranteed by a stable stem-loop structure generated immediately after their synthesis (Figure 1), together with their binding to chaperone proteins, such as the Sm family member Hfq [83], ProQ [84], and the prototype of its family CsrA [85], which stabilizes small RNA and promotes their binding with mRNA transcripts [84,86]. In detail, Hfq accelerates the formation of the small RNA/target RNA duplex in Gram-negative bacteria, modulating the decay, the transcription, or the translation of their RNA targets. Unlike *cis*-encoded small RNAs, *trans*-encoded small RNAs are transcribed from different genomic locations compared to their target genes (Figure 1). Like eukaryotic siRNAs, their binding to mRNA targets consists of an incomplete binding pair [74,79]. In addition, similar to eukaryotic miRNAs, they show more than one mRNA target and a binding site that comprise a highly conserved seed sequence of 6–8 contiguous nt (Figure 1). In general, the base pairing between small RNA and mRNA targets can activate or inhibit mRNA translation (by RNA Pol III) [87], mRNA stabilization (in example, GadY) [76], or mRNA degradation (for example, RyhB) [88]. Interactions between small RNAs and their ultimate targets may have either positive or negative effects on the expression of the regulated gene(s) [89,90]. The mechanisms by which *trans*-encoded small RNAs positively modulate the expression of their target genes include two processes. The first is characterized by the mRNA translational activation via structural rearrangements in the 5′-UTR of the mRNA transcripts, unmasking the ribosome-binding site (RBS) and leading to ribosomes access to mRNA transcripts for an efficient translation [66,91]. The second includes the enhancement of mRNA stability after binding to the mRNA 3′-UTR (small RNA–mRNA interactions) via undetermined mechanisms (Figure 1). On the other hand, a mechanism by which *trans*-encoded small RNAs can negatively modulate their mRNA targets, destabilizing and causing their degradation, is through the interaction and occlusion of RBSs in the mRNA transcripts, potentially stopping the target gene from being translated (Figure 1) [92].

The presence of positive and negative feedback loops suggests that they are part of a fine regulated circuit (Figure 1). An example of the negative feedback loop, similar to *cis*-encoded small RNAs, is *trans*-encoded small RNA RyhB acts as a sensor of iron levels in

bacteria. In particular, iron starvation promotes RyhB activation, which in turn suppresses the mRNA levels of iron-storage-related proteins and those of non-essential iron containing proteins, including ferritins, superoxide dismutase, succinate dehydrogenase, aconitase, and fumarase [93–95].

Recent findings indicate novel, alternative targets for *trans*-encoded small RNAs. Indeed, RNA chaperones interactome studies reveal that *trans*-encoded small RNAs prefer to bind different small RNAs, implying that they have more specialized roles [96,97]. For example, FasX, a small regulatory RNA from group A of *Streptococcus*, not only indirectly binds proteins and modulates transcription or translation, but can also increase streptokinase activity by increasing the stability of the ska mRNA transcript [70,98].

## 4. CRISPR

Many bacterial and archaeal genomes contain loci made up of regularly spaced repeats interspersed by other virus-derived DNA sequences, known as CRISPR-Cas proteins, which serve as mechanisms for bacteriophage defense [41]. The CRISPR consist of 28 to 37 base pairs of palindromic repeated elements, separated by spacers with an unique sequence or a similar length that represent the specificity of each CRISPR mechanism of action [99].

The CRISPR is an adaptive mechanism, where new spacers from phage genomic sequences can be integrated into bacteria upon viral stresses, conferring diverse resistances [100]. Therefore, CRISPR loci and Cas genes can provide high-speed and robust compatibility against viruses [101]. The first report of the CRISPR-Cas system was in 1987, where the CRISPR was described as an unusual genetic structure containing alternating repeated and non-repeated DNA sequences, but functional details were not described [102]. Since then, chromosome rearrangements, gene expression modulation, and DNA repair structures have all been proposed as functional roles [103]. However, similarities between some CRISPR spacer sequences and viral plasmid sequences contribute to the postulation of the hypothesis that the CRISPR may play a crucial role in adaptive immunity against foreign nucleic acids [104].

The increase of knowledge and data from the last decades on prokaryotes genome through NGS and, more recently, through single-cell sequencing association studies confirms that prokaryotes use the CRISPR system as an additional mechanism of defense against alien genetic infections. In particular, recent findings demonstrate that *Cas* genes are in closed proximity with CRISPR sites, critical for the CRISPR-mediated defense of bacteria. This versatile function of nucleases involves the identification and the attachment of specific DNA sequences and then the high-selective and high-specific cleavage of the newly formed double strand.

*CRISPR-Cas Classification, Structure, and Mechanism of Action*

CRISPR systems are divided into six categories that can be classified into two groups. The first CRISPR group, which has been the most extensively studied so far, includes categories (types) I, III, and IV and uses a set of different Cas proteins to exert their function. The second group has been recently identified, includes types II, V, and VI and uses only one Cas protein for their mechanism of action [105,106].

The CRISPR locus in the genome consists of three parts. Upstream to CRISPR loci, there is an AT-rich leader sequence of 500 bp long that carries promoter elements and adaptation elements for the *trans*-activation of CRISPR RNA (crRNA), responsible for integrating external genetic materials into the CRISPR sequences. The crRNA and the leader sequence are preceded by *Cas* genes. More than 40 different families of *Cas* genes have been identified to play important roles in crRNA biogenesis, production of spacers, and fragmentation of invasive DNA. Cas proteins require a protospacer sequence for their mechanism of action, which corresponds to the crRNA itself together with a protospacer adjacent motif (PAM) sequence [107]. The PAM sequence is upstream of the crRNA junction and on the complementary strand. Because this sequence is only on the invading genome, it is impossible to cut the host bacterial genome, for example, by Cas9 nuclease [108]. The

third part of the CRISPR locus consists of repeated and spacer sequences specifically to CRISPR systems. The repeated sequences are 25 to 50 bp in length, whereas the length and the spacer regions of non-repeated sequences are about 26 to 72 bp long [109]. They are arranged, so that the spacers are spaced between conserved repeated sequences. The repeated sequences are arranged in a palindromic way, meaning that the repeated sequences are identical and oppositely oriented compared to the opposite strand (Figure 2).

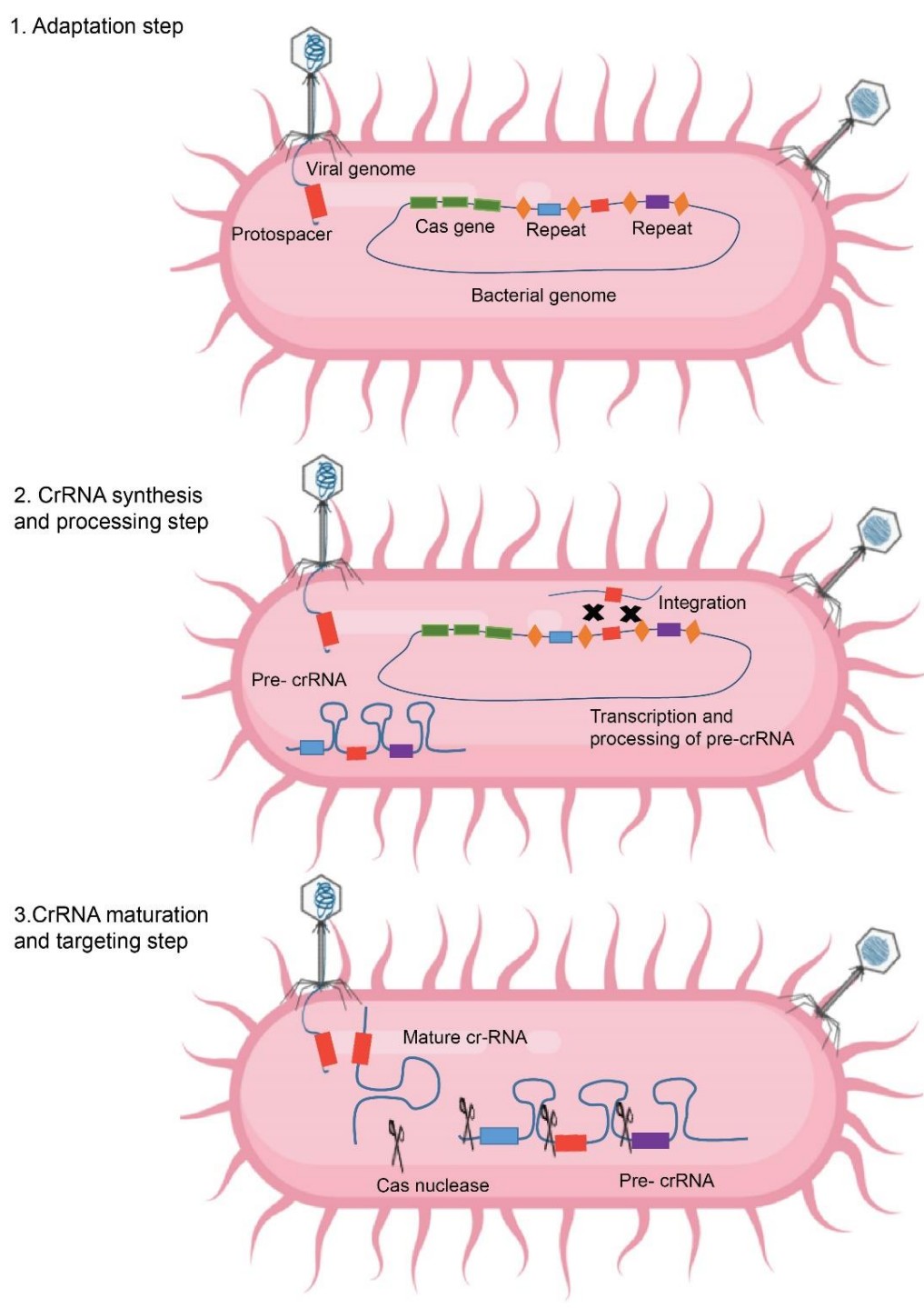

**Figure 2.** Clustered regularly interspaced short palindromic repeat's (CRISPR's) normal operation 1. Adaptation steps: the bacterial cell is invaded by phages, the phage genome enters the host cell, and

the protospacer fits into the CRISPR genome. 2. CRISPR RNA (CrRNA) synthesis and processing step: a single-stranded sequence of repeated and non-repeated (spacer) sequences, named pre-crRNA, is generated via the transcription from the CRISPR gene loci and eventually processed by Cas nuclease and converted to mature crRNA. 3. crRNA maturation and targeting step: pre-crRNA eventually processed by Cas nuclease and converted to mature crRNA, this molecule associated with Cas protein can target and degrade the invading DNA.

Here, we describe the CRISPR type II system as an example of the CRISPR-Cas9 systems, since it catches particular attention due to the known genetic engineering application and Nobel Prize won (see Section 6). The CRISPR-Cas type II relies on three components: the host RNase II, the small *trans*-activating RNA molecules (tracrRNA), and Cas9 proteins. The first step consists of the transcription of tracrRNA, which is complementary to the repeat sequences and contains three stem-loop hairpin structures that bind to a CRISPR locus-derived crRNA precursor, named pre-crRNA. The interaction between the tracrRNA and the pre-crRNA is stabilized by Cas9 [110], which helps the subsequent recognition and cleavage of the pre-crRNA by RNase III to generate a mature crRNA. The presence of the RNaseIII enzyme is a peculiarity of the CRISPR type II in bacteria and is absent in archaea [111]. Binding of Cas9 to the tracrRNA/crRNA complex induces a conformational change and the activation of Cas9, which in turn forms the active structure of the crRNA-guided endonuclease. In this system, after the first exposure to foreign generic elements, the invasive double-strand DNA is unfolded, and crRNA binds to a single-strand DNA generating an R-loop shape, which is transformed into small fragments by Cas nuclease. The fragments are inserted into spacers at the CRISPR locus. In response to a second infection from a virus or bacterium [112], this fragment is expressed and binds to its complementary strand in the invading genome. Cas9 is responsible for the double-strand break cutting of the invasive genome through its two catalytic sites, RuvC and HNH, upstream of protospacer sequences, named protospacer adjacent motifs (PAM, 2–5 nt long) [113].

## 5. Riboswitch

The ability of RNAs to modify their conformation upon binding increases the number of contacts with a given ligand and promotes the recognition of multiple targets, which is one of the advantages of RNA-based controls [114]. One of the most prominent examples is riboswitches, as they are *cis*-regulatory structured RNAs, playing important and widespread functions in the regulation of bacterial genes expression. Riboswitches can control the expression of downstream genes by straight response to small-molecule ligands [115]. The term "riboswitch" was coined to describe an RNA genetic switch that directly binds metabolites without additional cofactors, thus regulating gene expression by directly allosterically altering the structures of mRNA transcripts [116]. Most of prokaryotic riboswitches are situated in the 5′-UTR of mRNA transcripts that encode biosynthetic enzymes acting as metabolic transporters [117]. The sequences of riboswitches typically comprise two parts, i.e., the aptamer domain and the expression platform domain (Figure 3).

Aptamers are short RNA sequences that bind small target ligands with high specificity and affinity. Aptamer binding to the ligand compound stabilizes the aptamer structure and causes a conformational change in the aptamer tertiary structure. The aptamer–ligand interaction is translated in a biochemical response by the expression platform domain, leading to an inhibition or expression of selective target genes. The most common expression platform in bacteria is situated directly downstream of the aptamer domain. Hence, by the allosteric modulation of the 5′-UTR structure, the expression platform aims to attenuate the transcription or the translation of the target gene by forming a terminator or by sequestering the RBS. An alternative aptamer–ligand conformation can also increase the transcription or the translation of their target genes by forming anti-terminator or anti-sequester structures [118] (Figure 3). Like engineered aptamers, every natural aptamer within the riboswitch acts as a molecular sensor, recognizing its corresponding target molecule among a complex of other metabolites. The prokaryote aptamer replaces the

traditional protein component, which would otherwise act as a sensory function. As a result, aptamers monitor the transcription and/or translation processes [118,119].

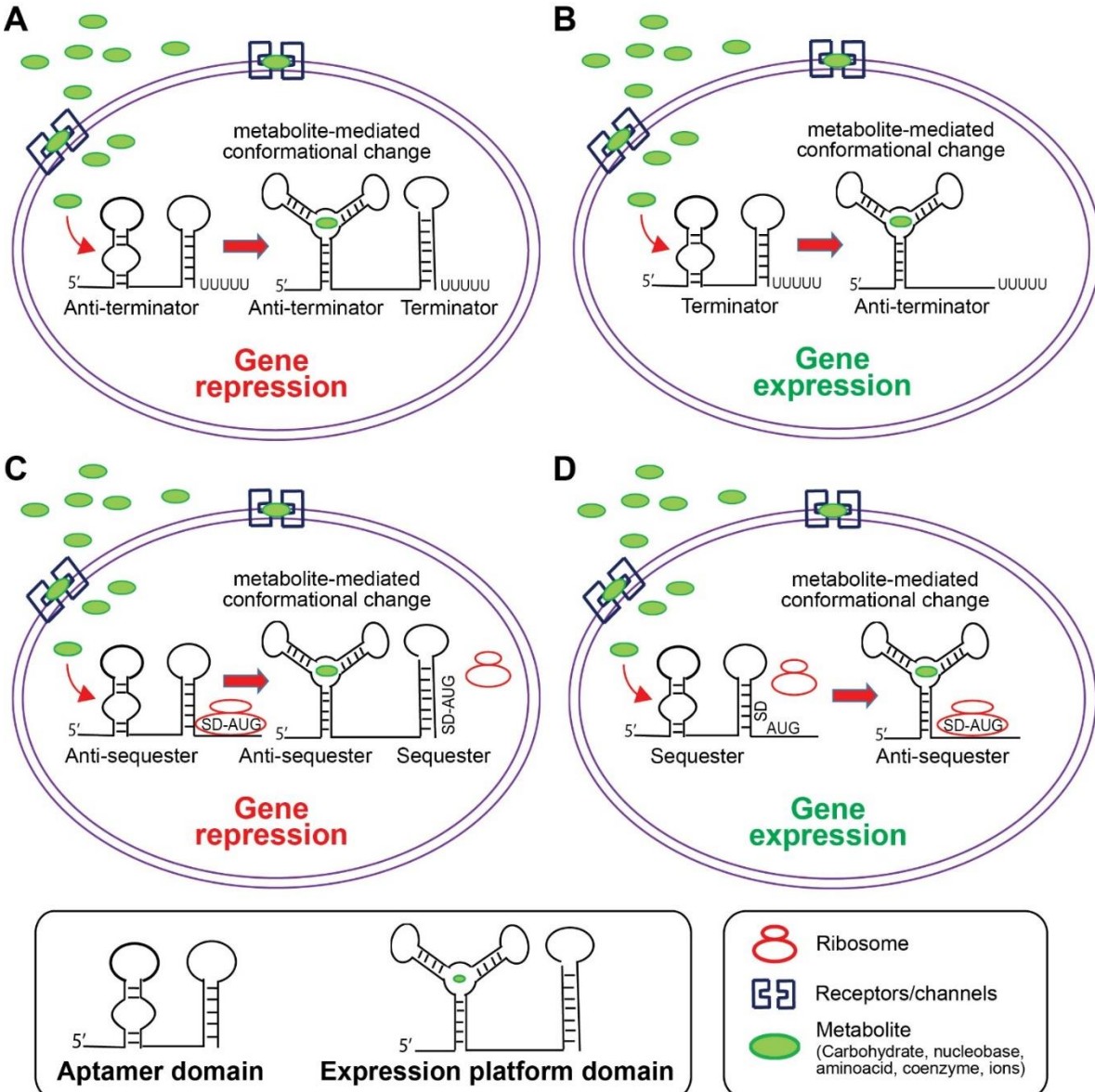

**Figure 3.** Gene regulation mediated by riboswitches. Bacterial riboswitches modulate their conformation in response to changes in the metabolite concentration and regulate the expression of downstream genes. Metabolite binding causes the repression or activation of downstream gene expression through transcription and/or translation. (**A**) Certain metabolite represses transcription through the formation of a specific step-loop structure (terminator) which stalls ribosome. (**B**) Certain metabolite activates transcription through the formation of a step-loop structure (anti-terminator), which allows ribosome to pass the mRNA. (**C**) Certain metabolite represses translation through the formation of a step-loop structure (sequester), which prevents ribosome's access to the Shine–Dalgarno (SD) sequence. (**D**) Certain metabolites activate translation through the formation of a step-loop structure (anti-sequester), which facilitates ribosome's access to the SD sequence.

### 5.1. Riboswitch Structural Classification

Based on their structural characteristics, riboswitches can be divided into two categories: (i) pseudoknoted and (ii) junctional riboswitches [115]. The RNA chain is mainly folded into a single-knot-like structure, consisting of two stem-loops, where the part of one loop is engaged in a base pairing with the second loop, according to the pseudoknoted riboswitches group. Such structures are seen, among others, in SAM-II, fluoride

riboswitches, and pre-queosine [120,121]. The ligand can bind to the junctional region of helices or to the groove of the helix, which is stabilized by pseudoknots. The junctional riboswitches are comprised of a central loop, performing a multi-helical junction role, and many radial helices. The number of helixes is changeable. For example, purine and thiamine pyrophosphate (TPP) riboswitches have a range of three helixes [122], whereas flavin mononucleotide (FNM) riboswitches comprise six helixes [108]. Evidence regarding the existence of riboswitches in eukaryotes is still poor, but a certain number of studies report that riboswitches or riboswitches-like RNAs exist in fungi, plants, and algae, in particular as TPP riboswitches [123–125].

### 5.2. Riboswitches Ligands and Regulatory Mechanisms

More than 20 notable riboswitch–ligand pairs have been discovered in recent decades, demonstrating that natural aptamers have the ability to selectively and tightly recognize different compounds [115] Additionally, natural expression platforms identify various mechanisms for ligand-dependent gene control by riboswitches in plants, fungi, and bacteria [125]. Although the ligand-binding site may interact with distal regions of the riboswitch, evidences suggest that the ligand-binding site is mostly localized within adjacent junctions [126].

Riboswitches are able to bind and sense cellular metabolites and ions. These include carbohydrates, coenzymes, nucleobases and derivatives, amino acids and derivatives, and ions, for example $Mg^{2+}$ [127,128]. Mechanistically, riboswitches act as "switchers" able to turn mRNA transcription on or off through selective bindings to the mRNA target. This mechanism is strictly dependent on riboswitches conformational changes. Indeed, riboswitches conformation switches between two mutually exclusive and thermodynamically stable conformations. Indeed, riboswitches are usually in a switch-off conformation, guaranteed by a minimum free-energy barrier that hinders spontaneous switching. When the concentration of a known small molecule reaches the binding threshold value, the small molecule interaction with the riboswitch aptamer domain promotes a conformational change in the second part of the riboswitch structure named expression platform. This binding promotes a riboswitch conformational change that passes through an intermediate, till final activated, state. Hence, the formation of riboswitches is realized in the following process: riboswitches-ligands complexes promote or inhibit the expression of their target genes by affecting their mRNA target transcription, degradation, translation, RNA interference, or splicing [126]. The most common mechanism of action of riboswitches resides in the modulation of transcription termination [129]. The transcription terminator, which is a stem followed by a chain of uridine residues [130], prevents RNA polymerase from completing the transcription process. The formation of the terminator is regulated by the formation of the aptamer domain as a result of metabolite binding, which creates a competitive anti-terminator [119]. Riboswitches use related structures to check the ribosome access to the RBS or Shine–Dalgarno (SD) sequence by stopping translation initiation in another general mechanism.

As mentioned before, the only riboswitch class discovered in fungi and plants is the TPP riboswitch [131–133], which regulates mRNA splicing. Although the mechanisms are not known, protein factors, such as RNases and the transcription termination factor Rho, might be involved in TPP riboswitch activity [134]. It seems that TPP riboswitches might hold anti-SD sequences in this type of mechanism to control both the transcription and the translation of their targets [135].

## 6. Novel Bacterial-Based Therapeutic Strategies against Genetic Diseases

Although the goal of this review is to summarize current and novel findings on functional RNAs in bacterial and archaea, we will briefly summarize current data underlying the potential applications of prokaryotic RNAs as novel therapeutic drugs against genetic diseases. Indeed, novel genetic engineering techniques have revolutionized how researchers are efficiently able to manipulate DNA molecules for therapeutic strategies.

Notably, genetic modification approaches were quite rudimental and overall ineffective till the 1970s [109]. Homologous recombination was the first effective gene-editing technique that successfully consents to manipulate genes to discover/study potential therapeutic targets. Accordingly, we will brief discuss the CRISPR-Cas9 and riboswitches system approaches and limitations as examples (Table 1).

**Table 1.** CRISPR- Cas9 and riboswitches therapeutic applications and limitations.

| Novel Bacterial-Based Therapeutic Strategies | | | |
|---|---|---|---|
| | **Beneficial Features** | **Clinical Applications** | **Limitations** |
| **CRISPR-Cas9** | Able to remove the dominant allele from the cell via non-homologous end joining (NHEJ) to correct errors during mitosis, avoiding subsequent potential genetic mutations and mitotic catastrophe | Limitations in the direct editing of a gene into the right cell type | Can be used for the genomic editing of diseases in which there is a need for editing a selective allele |
| | | Poor selectivity | Can be used for antibacterial therapies |
| **Riboswitches** | Identified riboswitches respond to ubiquitous and important metabolites and second messengers related with mRNAs-encoding proteins, fundamental for survival or against pathogens | Exhibit a limited selectivity for their target genes | |
| | Riboswitches-mediated small molecules recognition is through different mechanisms compared to eukaryotes, avoiding cross-species reactivity | Analog ligands still unknonw | |

### 6.1. CRISPR System as a Potential Target of Genetic Diseases: How Do We Do It "Right"?

The CRISPR-Cas9 system has gained particular attention in the past decades due to its utilization as a tool to literally change the DNA of animals, plants, and micro-organisms with an extremely high precision. Accordingly, Charpentier and Doudna awarded the Nobel Prize in Chemistry in 2020 for the application of the CRISPR-Cas9 system for DNA genetic engineering.

Although the CRISPR is widely used as a research tool, it can also be used as a tool to target genetic diseases. In general, the CRISPR technique can be used for the genomic editing of diseases, in which there is a need for editing a selective allele [136]. Moreover, the CRISPR-Cas technology enables researchers to target multiple genes simultaneously to understand the pathological processes that occur due to the disruption of several genes. Examples of these pathological processes are found in cancers [137].

If the observed disease results from a genetic defect and produces a defective molecule within a cell, the CRISPR has the potential to remove the dominant allele from the cell, a mechanism called non-homologous end joining (NHEJ). The NHEJ is an intrinsic mechanism adopted in eukaryotic cells during cell division and angiogenesis to correct errors during mitosis, avoiding subsequent potential genetic mutations and mitotic catastrophe. In prokaryotes, if the disease results from the loss of function of a gene, the defective allele is replaced with a healthy allele by homology-directed repair (HDR). Each Cas9 system identifies its own PAM sequence [136]. In the NHEJ method, proteins involve binding two strands of DNA, either directly or by using the deletion and addition of nucleotides, and this mechanism does not need DNA as a template. In opposite, HDR uses DNA as a template for accurate repair [105].

Another noteworthy application of the CRISPR is its use in the treatment of single-gene genetic abnormalities. Examples of these diseases include Duchenne's muscular dystrophy (DMD), cystic fibrosis (CF), and hemoglobinopathies [138]. Schwank and

colleagues investigated and treated CF using the CRISPR-Cas9 system. In detail, authors successfully modified the Cystic fibrosis transmembrane conductance regulator (CFTR) mutation, which characterizes CF, in intestinal stem cells obtained from two adults with CF. They showed that through a single-vector-induced correction of the mutated gene, the function of the CFTR-transmembrane vector was restored, thereby compensating and eliminating somehow the disease [139].

Another disease examined by the CRISPR system is DMD. Tabebordbar and colleagues recently used the CRISPR to retrieve dystrophin gene expression in a DMD mouse model. They did so by deleting the mutated exon of the original gene, which resulted in the shortening of the gene but still in the production of a functional protein. The work from Tabebordbar and colleagues indicates that that muscle function is significantly improved. Most important, the results of CRISPR treatment did not disappear over time [140].

It has been shown that the injection of this system into the zygote or during early blastocysts allows the cell genome to be modified [141]. Therefore, the CRISPR might be used to make permanent changes and eliminate inherited genetic diseases. Accordingly, Wu and colleagues used this approach for the first time to modify genetic diseases in mouse embryos and corrected the *Crygc* gene mutation that causes cataracts [142].

However, some limitations exist in the CRISPR system. For example, it still arises the question of how to direct the editing of a gene into the right cell type. The question is then: how do we do it "right"? The right selective gene-editing process into the right cell would require the right vector. Notably, one of the vector-associated gene therapies currently tested relies on adeno-associated virus (AAV) as a vector for gene therapies, but this carrier is small and may not transmit a sufficient Cas9 gene [109]. Another limitation is the lack of selectivity, which results in the modification of non-target genes. Notably, an unwanted genome editing can have irreversible effects on patients [143].

*6.2. Riboswitches as Novel Targets in Antibacterial Therapies*

Bacteria have the intrinsic ability to constantly adapt to and develop resistances to most of well-known drugs [144–146], leading to antibiotic resistance. Antibiotics can bind to a variety of targets, including enzymes involved in the biosynthesis of folic acid [147], DNA topoisomerase enzymes [148], and ribosomes [149] and in the biosynthesis of cell membrane components [150]. To avoid bacterial antibiotic resistance, novel pluripotent or multitarget antibacterial agents are required. Unfortunately, identifying critical processes and pathways that are sufficiently broad and conserved among all prokaryotes to allow the development of novel potential antibiotics with a broad spectrum of action is difficult [151]. The presence of riboswitches in bacteria and their capability to accurately identify different molecules make riboswitches potential candidates to generate novel and efficient antibacterial therapies. The relevance of riboswitches as a therapeutic target is underlined by the most emerging problem residing in currently available antibacterial treatments, since several bacteria are resistant to conventional antibiotics [126].

Riboswitches have been engaged as potential targets to develop drugs for three main reasons. Although eukaryotic and prokaryotic riboswitches have the potential to recognize the same small molecules, riboswitches are more abundant in prokaryotes and recognize their targets through different, distinct mechanisms. Hence, despite the fact that the target might be similar, prokaryotic-riboswitch-based drugs can be selective and lack non-specific side effects. Second, except for TPP riboswitches, other known riboswitches predominantly exist only in bacteria, not eukaryotes. If eukaryotes use riboswitches, these will probably be different from those in bacteria, minimizing the cross-reactivity of bacterial riboswitch-targeted ligands [152]. Third, identified riboswitches respond to ubiquitous and important metabolites and second messengers and are frequently related with mRNAs-encoding proteins, fundamental for survival or pathogenesis. Notably, riboswitch connection with a selective target gene is extremely conserved across phylogeny [153], underlining their physiologic importance.

There are critical limitations that should be underlined before considering riboswitches as potential targets to develop efficient therapeutic drugs. First, although riboswitches have riboswitch RNA elements, they exhibit a limited selectivity for their target genes [154,155]. Second, to generate effective anti-riboswitches small molecules, it is necessary to discover analog ligands with the potential to induce an effective switch of riboswitch conformation that could permanently influence riboswitches, even in the absence of native ligands, to prolong a potential beneficial effect [126]. Since the normal human gut microbiome comprises bacterial and archaeal colonies, riboswitches inhibition may cause a diseased condition in humans via disturbance in gut microbiome composition. Therefore, synthetic anti-riboswitches should be able to recognize pathogenic, and not gut, microbiota. For example, in *Bacillus subtilis*, roseoflavin is an analog ligand that binds to a flavin mononucleotide riboswitch and inhibits the growth of bacteria by suppressing the biosynthesis and the transport of the riboflavin [156]. However, bacteria have a lot of ncRNAs, which could lead to new RNA-based targets for antibacterial drugs development [151,157].

## 7. Conclusions

In the last decades, it has been displayed that the "silent" genome is involved in the fine-tune regulation of several cellular mechanisms, including gene expression and protein assembling. In this regard, hundreds of RNAs have been identified, each playing a specific role in regulating cellular processes and gene expression. The processing and destruction of mRNAs plays an essential role in cell life and the RNA-based regulation of genes. In addition to the RNA-based regulation of gene expression through mRNA targeting, some of these RNAs can directly bind and modulate proteins function. Genome-wide analysis opens up new aspects of prokaryotic biogenesis and translational research. Indeed, data here summarized on prokaryotic small RNAs, such as miRNA-like RNAs, riboswitches, and the CRISPR-Cas system support the hypothesis that they can be used to generate more highly selective and highly efficient antibacterial drugs. Overall, Future NGS and genome-wide studies will help us to dissect the molecular mechanisms involved in prokaryotes gene regulation, those conserved or undergoing evolutionary development and those with a certain homology in prokaryotes and eukaryotes (for example, conserved epigenetic molecules, from which ncRNAs are partly come). Next, studies on the prokaryotic genome should examine and understand the function of these RNAs more in detail, focusing on their binding and pairing with their mRNA targets. Although it is still challenging to predict their mechanism of binding and their binding sites on mRNA targets in silico, it might significantly help to dissect all other potential roles exerted by prokaryotic small RNA molecules potentially applicable for drug design.

**Author Contributions:** The authors confirm the contributions to the paper as follows: study conception and design, Z.S., Z.A.K., A.M. and L.N.; data collection, E.S.-F., S.T. and Z.A.K.; analysis and interpretation of results, Z.A.K., L.N. and C.W.; draft manuscript preparation, M.T.-A., S.H.K., P.M., Z.A.K. and L.N.; supervision and revision of the manuscript, C.W., A.M. and L.N. All authors have read and agreed to the published version of the manuscript.

**Funding:** This research received no external funding.

**Institutional Review Board Statement:** Not applicable.

**Informed Consent Statement:** Not applicable.

**Data Availability Statement:** All the related data are deposited in the manuscript.

**Conflicts of Interest:** The authors declare there are no conflict of interest.

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
