# Peer review of "Insights into the Function of Regulatory RNAs in Bacteria and Archaea"

_2673-8937, doi:10.3390/ijtm1030024_

Round 1

Reviewer 1 Report

In this manuscript, the authors reviewed the function of various types of the regulatory RNA in bacteria and archaea, such as riboswitch’s detection and response to available nutrients within the cell through the regulation of gene expression. I guess the topic that describes the known mechanism and regulatory roles of non-coding RNAs is interesting and need to be reviewed in a paper, but in this paper, not only the regulatory mechanisms of non-coding RNAs in bacteria and archaea but also the therapeutic usage of the mechanism in human genetic disease especially in the CRISPR section unexpectedly. And this manuscript should go through a grammar correction and editing process before publication. Here I present some parts that need to be corrected, but there may be more things to be fixed.

Line 30, a semicolon should be changed into a comma or These -> these.

Line 88, kb -> Kb.

Line 118, no space after a comma.

Lines 129 and 302, double space in front of the reference number.

Lines 136-137, 267, 275, and 509, a comma in front of the reference number.

Line 140, Bacterial -> bacterial.

Lines 159, 204, and 291, double space after the reference number.

Lines 160 and 611, reference numbers should be enclosed in parentheses.

Line 321, Streptococcus -> Streptococcus.

And I could not find Figure 1, described in lines 420-422.

Author Response

In this manuscript, the authors reviewed the function of various types of the regulatory RNA in bacteria and archaea, such as riboswitch’s detection and response to available nutrients within the cell through the regulation of gene expression. I guess the topic that describes the known mechanism and regulatory roles of non-coding RNAs is interesting and need to be reviewed in a paper, but in this paper, not only the regulatory mechanisms of non-coding RNAs in bacteria and archaea but also the therapeutic usage of the mechanism in human genetic disease especially in the CRISPR section unexpectedly. And this manuscript should go through a grammar correction and editing process before publication. Here I present some parts that need to be corrected, but there may be more things to be fixed.

We thank the Reviewer for the constructive comments, which help us to significantly improve our Review. The manuscript has been carefully checked and significantly modified to correct all English grammar errors, as well as to ameliorate all parts that were not properly described. We are happy that the Reviewer agree with us about the importance to summarize and discuss current knowledges about bacterial non-coding RNAs. Indeed, this topic got particular attention in the research committee during the last years, especially regarding the CRISPR/Cas system. Indeed, the system is now used in a wide spectrum of fields, which include the generation of transgenic animals for research purposes and of novel therapeutic approaches. Accordingly, Emmanuelle Charpentier and Jennifer Dudna have been awarded the 2020 Nobel Prize in Chemistry for their work on CRISPR-Cas9 as a method to edit DNA. Remarkably, only eight years after its inception, clinical trials are underway to test whether CRISPR-Cas9 may be used to treat inherited diseases such as β-thalassemia or sickle cell disease (Doudna JA. The promise and challenge of therapeutic genome editing. Nature. 2020;578:229–236. doi: 10.1038/s41586-020-1978-5.). Hence, we decided to dedicate a section at the future therapeutic perspectives based on bacteria-like RNAs, including CRISPR and Riboswitches. We understand that this part should be discussed at the end of the Review. Therefore, we now dedicated a new Pragraph (paragraph 6) in our Review. We believe that this part can be considered a future perspective and would better complete current knowledges on bacterial ncRNAs. In addition, we have also included some limitations that should be considered (new Table 1).

Line 30, a semicolon should be changed into a comma or These -> these.

Line 88, kb -> Kb.

Line 118, no space after a comma.

Lines 129 and 302, double space in front of the reference number.

Lines 136-137, 267, 275, and 509, a comma in front of the reference number.

Line 140, Bacterial -> bacterial.

Lines 159, 204, and 291, double space after the reference number.

Lines 160 and 611, reference numbers should be enclosed in parentheses.

Line 321, Streptococcus -> Streptococcus.

All minor concerns have been addressed

And I could not find Figure 1, described in lines 420-422.

We have now included 3 new figures and a Table.

Reviewer 2 Report

The review article “Insights the function of regulatory RNAs in bacteria and archaea” by Fard et al. describes the roles of non-coding regulatory RNAs, focusing on small RNAs and riboswitches’ regulatory functions. The topic is important, and the manuscript is well written. However, I wish the authors address the following comments. The manuscript must be improved.

Major comments:
(1)
I strongly recommend that you should use a lot of figures (diagrams or cartoons) for better understanding in your review. At least five figures (or more) that explain the content are required. You may use the figures in the references via the “rights and permissions” procedure.

(2)
Line 422, Where is “Figure 1”? Figure 1 is not shown.

Minor comments:
Some typo in the manuscript should be corrected. Please check all typos again in your manuscript.
e.g.
Line 123, “in vivo” should be corrected to “in vivo” (italic).
Line 160, “et al.” should be corrected to “et al.” (italic).
Line 217, “C. elegans” should be corrected to “C. elegans” (italic).
Lines 170-171, 238, 272, 346, “E. Coli” should be corrected to “E. coli” (italic).
Lines 242, 302, 305, 604, “via” should be corrected to “via” (italic).
Line 539, “Mg2+” should be corrected to “Mg2+” (superscript).
In your manuscript, “cis” and “trans” should be corrected to italic character.

Author Response

The review article “Insights the function of regulatory RNAs in bacteria and archaea” by Fard et al. describes the roles of non-coding regulatory RNAs, focusing on small RNAs and riboswitches’ regulatory functions. The topic is important, and the manuscript is well written. However, I wish the authors address the following comments. The manuscript must be improved.

Major comments:
(1) I strongly recommend that you should use a lot of figures (diagrams or cartoons) for better understanding in your review. At least five figures (or more) that explain the content are required. You may use the figures in the references via the “rights and permissions” procedure.

We thank the Reviewer to appreciate our work. The manuscript has been carefully checked and significantly modified to ameliorate all parts that were not properly described. We agree that additional figures can help the readers for better understandin the Review. Accordingly, we generated 3 new figures describing: the biogenesis and function of cis- and trans-encoded small RNAs (Figure 1), CRISPR (Figure 2), and Riboswitches (Figure 3). Moreover, since we believe that is important to discuss the therapeutic strategies currently adopted using bacterial-like RNAs as well as current limitations, we have now included a table at the end of the new paragraph 6 (Table 1).

(2) Line 422, Where is “Figure 1”? Figure 1 is not shown.

The figure has been included

Minor comments:
Some typo in the manuscript should be corrected. Please check all typos again in your manuscript.
e.g.
Line 123, “in vivo” should be corrected to “in vivo” (italic).
Line 160, “et al.” should be corrected to “et al.” (italic).
Line 217, “C. elegans” should be corrected to “C. elegans” (italic).
Lines 170-171, 238, 272, 346, “E. Coli” should be corrected to “E. coli” (italic).
Lines 242, 302, 305, 604, “via” should be corrected to “via” (italic).
Line 539, “Mg2+” should be corrected to “Mg2+” (superscript).
In your manuscript, “cis” and “trans” should be corrected to italic character.

All minor concerns have been addressed

Round 2

Reviewer 1 Report

Minor point: within Figure 2 'Adaption' should be 'Adaptation'.

Author Response

We thank the reviewer for the comment.

We corrected the Figure 2 accordingly

Reviewer 2 Report

All comments were addressed carefully. The manuscript was refined.
I am satisfied with the revision and recommend for publication as is.

Author Response

We thank the reviewer for the positive comments and to appreciate the revised form of our Review